# JUDGING THE JUDGES: EVALUATING ALIGNMENT AND VULNERABILITIES IN LLMS-AS-JUDGES

## ABSTRACT

Offering a promising solution to the scalability challenges associated with human evaluation, the *LLM-as-a-judge* paradigm is rapidly gaining traction as an approach to evaluating large language models (LLMs). However, there are still many open questions about the strengths and weaknesses of this paradigm, and what potential biases it may hold. In this paper, we present a comprehensive study of the performance of various LLMs acting as judges[1], focusing on a clean scenario in which inter-human agreement is high. Investigating thirteen *judge models* of different model sizes and families, judging answers of nine different '*exam-taker models*' – both base and instruction-tuned – we find that only the best (and largest) models achieve reasonable alignment with humans. However, they are still quite far behind inter-human agreement and their assigned scores may still differ with up to 5 points from human-assigned scores. In terms of their ranking of the nine exam-taker models, instead, also smaller models and even the lexical metric `contains` may provide a reasonable signal. Through error analysis and other studies, we identify vulnerabilities in judge models, such as their sensitivity to prompt complexity and length, and a tendency toward leniency. The fact that even the best judges differ from humans in this comparatively simple setup suggest that caution may be wise when using judges in more complex setups. Lastly, our research rediscovers the importance of using alignment metrics beyond simple percent alignment, showing that judges with high percent agreement can still assign vastly different scores.

## 1 INTRODUCTION

Over the last few years, large language models (LLMs) have demonstrated remarkable capabilities across various domains (Radford et al., 2019; Brown et al., 2020; Achiam et al., 2023; AI@Meta, 2024, i.a.). As more and more new LLMs with different architectures and training methods continue to be released and their capabilities expand, accurately evaluating their performance and limitations becomes increasingly challenging (Zheng et al., 2024; Ohmer et al., 2024; Benchekroun et al., 2023; Madaan et al., 2024). The empirical evaluation of LLMs is particularly difficult due to the diversity of their outputs and the wide range of tasks they are used for (Zhang et al., 2024; Li et al., 2023a).

To evaluate LLMs, various methods have been proposed, typically falling into one of two broad categories. First, benchmarks such as MMLU (Hendrycks et al., 2021), TruthfulQA (Lin et al., 2021), or GSM8K (Cobbe et al., 2021) are used to evaluate specific capabilities of LLMs in an automated manner. Additionally, leaderboards like Chatbot Arena (Chiang et al., 2024) and Open LLM Leaderboard (Beeching et al., 2023) assign ranks to models considering pair-wise rankings of LLM outputs, done by humans or, in some cases, automated evaluation methods. Since both strategies involve evaluating free-form text responses generated by

---

[1]Source code is available in the supplementary material.

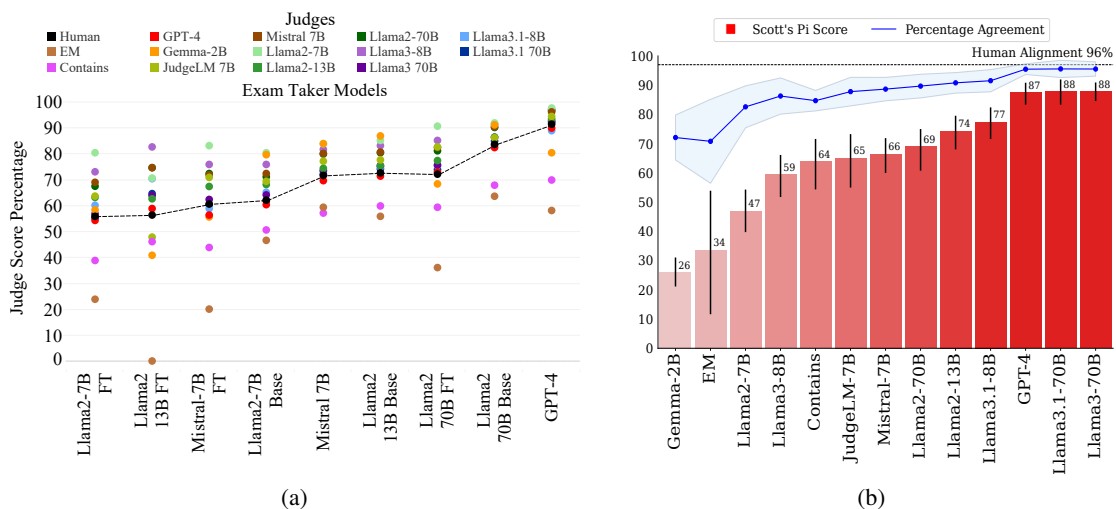

Figure 1: **Average scores assigned by judge models and alignment with human judges.** (a) Scores assigned to all exam-taker models by the various judge models. (b) Average percent agreement (blue line) and Scott's pi scores (red bars) of judge models with human judges (black line). Error bars annotate standard deviation across exam-taker models. `Llama3 70B`, `Llama3.1 70B` and `GPT-4 Turbo` have Scott's pi coefficient that are indicative of excellent alignment, but are still well below the human alignment score.

the LLMs, even in the first case, evaluating the responses is often just as challenging as generating them (see e.g. Chang et al., 2023; Bavaresco et al., 2024).

One proposed solution to this problem is to use multiple-choice question (MCQ) benchmarks such as MMLU, and compare the log-probabilities of the potential answers rather than evaluating the generated answer directly. However, the MCQ paradigm limits the range of abilities that can be evaluated, and the setup increasingly diverges from how LLMs are used in practice. Alternatively, the use of lexical matching methods such as exact match (EM) or n-gram overlap to evaluate the responses are practical and cost-efficient approaches, but are susceptible to false negatives and often fail to adequately distinguish between responses with subtle differences that change their semantic meaning. This issue is exacerbated when evaluating instruction-tuned "chat" models that are fine-tuned to carry out conversations with humans in natural language, since their responses tend to be more verbose (Saito et al., 2023; Renze & Guven, 2024). For these reasons, human evaluation remains the gold standard for evaluating LLM responses.

However, human evaluation is expensive, time-consuming, and often impractical in many use cases. As a result, it has increasingly become common practice to evaluate LLM responses using another LLM as a judge model (Lin et al., 2021; Islam et al., 2023; Chiang & Lee, 2023; Liusie et al., 2024). While there are promises of alignment between LLM judges and humans (Sottana et al., 2023; Zheng et al., 2024), there are also many open questions about the strengths and weaknesses of the paradigm. In this work, we study the properties of LLMs as judges, comparing them with humans and automated evaluation methods. Contrary to prior work, we focus on a clean scenario in which human alignment is very high, allowing us to distinguish ambiguity and subjectivity in the task itself from potential issues with the judge models. Using the knowledge benchmark TriviaQA (Joshi et al., 2017) as our playground, we investigate how thirteen different *judge models* with varying architectures and sizes judge nine different *exam-taker models*. Our main findings are:

- **Even in clean and straightforward setups, only the best models have high alignment scores**. Out of the thirteen judge models we considered, only `GPT-4 Turbo`, `Llama-3.1 70B` and `Llama-3 70B` showed very high alignment with humans. Also for those judges, though, alignment is still well behind the human alignment coefficient for the task (Figure 1).
- **Scott's $\pi$ distinguishes judges better than percent alignment**. In terms of percent alignment, judges are rarely discriminable, while Scott's $\pi$ provides a more informative signal. In some cases, high percent agreement can still give scores that differ 10-20 points from the human-assigned scores (Figure 2).
- **Also Scott's $\pi$ is not all telling**. While `GPT-4 Turbo` and `Llama-3` both have alignment scores that are considered excellent, their scores still differ up to 5 points from human-assigned scores. Furthermore, when it comes to *discriminating* different exam-taker models, their results are comparable to alternative cheaper approaches such as `Mistral 7B` and `contains`, which have much lower alignment scores but more consistent biases (Figure 3).

Through detailed analysis (§ 5), we uncover additional insights into judge performance. Improved alignment appears to be driven by improved recall rates and reduced false negatives. However, judge models struggle with under-specified answers and tend to be lenient, affecting their evaluation consistency. They are also sensitive to the length and quality of prompts. And, surprisingly, even when the judge models are asked to evaluate an answer matching verbatim with a reference answer, many judge models still sometimes fail to evaluate it correctly.

Overall, our work showcases the strengths of the LLM-as-a-judge paradigm while also highlighting the need for caution against overreliance on alignment metrics, even in cases where they are high. Through error analysis, we also highlight several common failure cases that require attention. With this, we aim to contribute to a better general understanding of what is now becoming a mainstream paradigm for evaluating LLMs.

## 2 RELATED WORK

Various recent studies have used or considered using LLMs as judges for tasks such as evaluating story generation (Chiang & Lee, 2023), retrieval-augmented generation (Es et al., 2023), visual QA (Mañas et al., 2024), code comprehension (Zhiqiang et al., 2023), multilingual evaluation (Hada et al., 2023) and more general open-ended tasks (Zheng et al., 2024). Zhang et al. (2024) and Sottana et al. (2023) propose ways to standardise LLM evaluations and the role that judge models might play in such solutions. Several studies have demonstrated that state-of-the-art LLMs such as GPT-4 Turbo exhibit high alignment with human judgments (Sottana et al., 2023; Zheng et al., 2024), though others also illustrate that the paradigm is not yet without faults. Zeng et al. (2023) propose a benchmark for evaluating the performance of LLMs as judges, and other approaches have been proposed to improve LLM judges such that they are aligned well with humans (Shankar et al., 2024; Zhu et al., 2023).

Despite promising results in various settings, judge models still suffer from known issues of current LLMs such as hallucinations and factual errors (Ye et al., 2023; Turpin et al., 2023) and difficulty in following complex instructions (Li et al., 2023b; He et al., 2024). Furthermore, various studies have reported challenges such as position bias (Pezeshkpour & Hruschka, 2023; Zheng et al., 2023; Wang et al., 2023), verbosity bias (Saito et al., 2023) in their preferences, confusing evaluation criteria (Hu et al., 2024), or focusing more on the style and grammar compared to factuality (Wu & Aji, 2023). Recently, Liusie et al. (2024) have shown that LLMs perform better in comparative assessment compared to absolute scoring, which can be used for reliably measuring the relative performance of models (Liu et al., 2024) and creating classifiers for pairwise grading (Huang et al., 2024).

We follow up on this line of work and investigate the strengths and weaknesses of LLMs as judges. Unlike most prior work, we do not focus on pairwise comparisons of LLM outputs on open-ended tasks, but on comparisons of LLM outputs and reference answers. Since human alignment is high in this setting, this

Table 1: **Exam-taker models and judge models** We consider a wide variety of exam-taker models and judge models; to get a in-depth overview of their abilities, we consider exam-taker models of various sizes & types.

| Exam-taker models (base & instruction-tuned) | Llama-2 (7B, 13B and 70 B), Mistral 7B, GPT-4 Turbo |
|---|---|
| Judge models (instruction-tuned) | Llama-2 (7B, 13B, 70B), Llama-3 (8B, 70B), Llama-3.1 (8B, 70B), Gemma 2B, Mistral 7B, JudgeLM 7B, GPT-4 Turbo |
| Judge models (lexical) | Exact Match (EM), Contains |

provides a clean playground to study the strengths and weaknesses of LLMs in detail. We also extend previous work by considering more LLMs, both as judges and LLMs to be evaluated.

## 3 METHODOLOGY

To evaluate the strengths and weaknesses of the LLM-as-a-judge paradigm, we focus on a comparatively controlled setup, in which judge models assess answers of exam-taker models on the knowledge benchmark TriviaQA (Joshi et al., 2017). With this methodological design, it is possible to focus on the abilities of the judges in isolation, without having to address human disagreement and error at the same time. In this section, we elaborate the main aspects of our methodology.

**Evaluation data**  As our testbed, we use the TriviaQA dataset (Joshi et al., 2017), consisting of 95K question-answer pairs sourced from 14 trivia and quiz league websites. Each question in the train and validation set is annotated with a list of short answers containing a minimal set of facts and evidence documents collected from Wikipedia and the Web. For our experiments, we use the validation set of the *unfiltered* partition of the benchmark, using the short answers as reference answers. We use the training set for few-shot examples. Since experiments require manual annotation of the exam-taker model responses, we use a random sample of 400 questions from the dataset. In Appendix I, we show with a bootstrapping test that this sample size has low variance for our main result. Through experiments described in § 3, we establish that humans have high agreement on judgements of answers given to the questions in the benchmark.

**Exam-taker models**  To understand the strengths and weaknesses of different judges, we consider answers of pre-trained (base) and instruction-tuned (chat) 'exam-taker models' across a wide variety of model sizes. In particular, we consider Llama-2 (Touvron et al., 2023) in 7B, 13B, and 70B parameter sizes for both base and chat versions, Mistral 7B (Jiang et al., 2023) base and chat versions, and GPT-4 Turbo[2] (Achiam et al., 2023) as the exam-taker models. The prompts for the exam-taker models contain five few-shot examples of (question, answer) pairs from the TriviaQA training set. The prompts for the instruction-tuned models additionally include a command signaling the model to answer the given question in a succinct manner similar to the provided examples. The prompts are provided in Appendix D.

**Judge models**  To get a comprehensive view of the strengths and weaknesses of judge models across different model sizes and architectures, we use instruction-tuned versions of Llama-2 (Touvron et al., 2023) in 7B, 13B, and 70B sizes, Llama-3 (AI@Meta, 2024) in 8B and 70B sizes, Llama-3.1 (Dubey et al., 2024) in 8B and 70B sizes, Mistral 7B (Jiang et al., 2023), GPT-4 Turbo (Achiam et al., 2023), Gemma 2B (Gemma Team et al., 2024), and JudgeLM 7B (Zhu et al., 2023) as judges. To maintain parity with human and judge evaluation, judge prompts were built from human guidelines in Appendix G. The judges are instructed to respond with only a single word, "correct" or "incorrect". An overview of

---

[2]Accessed via the OpenAI API between Mar 19th, 2024 and Sep 20, 2024.

all exam-taker models and judge models is shown in Table 1. For ease of reading, the `judge models` are depicted in a different font than the exam-taker models.

**Baselines**   As baselines, we use two commonly used lexical evaluation techniques – exact match (`EM`) and contains match (`contains`). For `EM`, a response is considered correct if the response exactly matches one of the reference answers for the given question. For `contains`, an answer is considered correct if at least one of the reference answers is a sub-string of the response string. Both EM and contains match are computed in a case-insensitive manner.

**Alignment**   We use two metrics to quantify alignment between judges: percent agreement and Scott's Pi coefficient (Scott, 1955).[3] Percent agreement expresses a simple percentage of the samples on which two annotators agree. Scott's Pi, denoted as Scott's $\pi$, is an alignment metric that corrects for chance agreement between two annotators and is considered to provide a more robust measure of alignment. Details about the computation of both metrics are given in Appendix F.

**Human judgements**   As a ground-truth assessment, we obtain human annotations for each exam-taker model answer. The inter-human alignment is calculated between three human judges using the answers to 1200 randomly sampled questions answers; the human guidelines can be found in Appendix G. We then determine collective "Human Judgment" through a majority vote. The average alignment among human evaluators with the majority vote had a Scott's $\pi$ of $96.2 \pm 1.07$,[4] and the average percent agreement was $98.52\% \pm 0.42\%$. The details of this experiment are mentioned in Appendix A. Given this near-perfect alignment score, we consider only one human evaluator per sample for the rest of our experiments, to reduce the overall cost of human annotations. The set of questions for which we obtain human annotations is identical for each exam-taker model.

## 4   RESULTS

In this section we discuss our main results, primarily focusing on the relationship between evaluations by various judge models and human evaluations (§ 4.1), and how that impacts their usability (§ 4.2). To do so, we evaluate their alignment with human judgment and assess how differently they rank the nine exam-taker models compared to humans. In Section 5, we further analyse their precision and recall to further investigate the types of errors that can be made by various judge models. Details about compute requirements and others costs for experiments are given in Appendix H.

### 4.1   ALIGNMENT BETWEEN JUDGE MODELS AND HUMANS

We start by computing Scott's $\pi$ scores and percent agreement between the evaluations of each judge model and the human annotators. We show the result in Figure 1. We observe that percent alignment is high for virtually all models, with the exception of `Gemma 2B` and `EM`. Scott's $\pi$, on the other hand, has low values for most models, though its value is in the high 80s for `Llama-3 70B`, `Llama-3.1 70B` and `GPT-4 Turbo`. Nevertheless, there still is a significant disparity between human judgment and judge models: the best scoring judge, `Llama-3 70B`, is 8 points behind human judgment. Notably, `EM` has the most variance in alignment, while `Gemma 2B` has the lowest alignment amongst all judges.

In most cases, we observe that Scott's $\pi$ and percent agreement are following the same trend, with the exception of the values for `Gemma 2B` and `EM`. `Gemma 2B` shows higher percent agreement compared to `EM`, yet it yields the lowest Scott's $\pi$ score within the ensemble. For the percent agreement of judge models, we note a 26-point difference between human judgment and EM, while Scott's $\pi$ exhibits a more substantial 64-point gap. This is also visible in the general decline of alignment scores: while `Llama-3 8B` has a Scott's

---

[3]In an earlier version of this paper, we used Cohen's kappa (Cohen, 1960) to measure alignment. It has since come to our attention that – despite it's widespread use – this metric has some well-documented theoretical issues (e.g. Pontius & Millones, 2011; Chicco et al., 2021). For the interested reader, we elaborate on these issues in Appendix B.

[4]coefficient scaled by 100 for easier comparison with percent alignment.

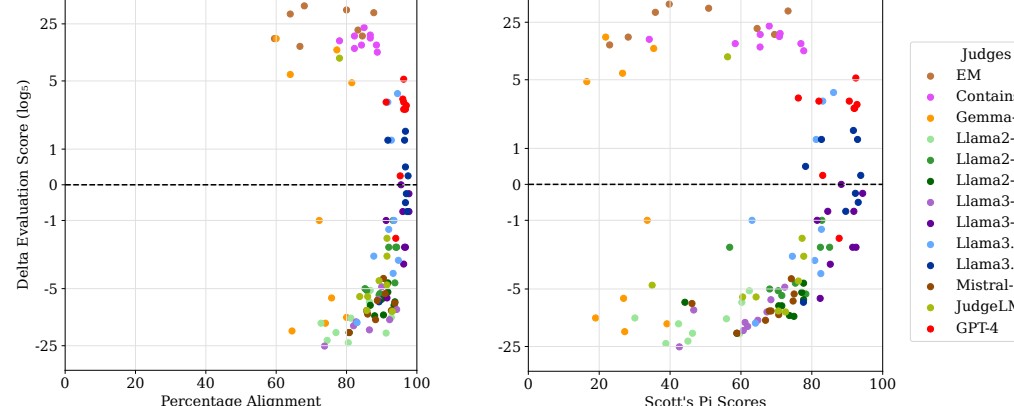

Figure 2: **Difference with human evaluation scores versus alignment metric.** The delta evaluation score is the difference between the judge and the human score; y-axes are in log scale. Percent alignment (left) shows a very skewwed distribution, making it difficult to distinguish models. Scott's $\pi$ (left) provides a clearer difference between models, and is more indicative of deviation of the gold score.

$\pi$ score of only 59, its percent agreement is still well above 80%. Overall, Scott's $\pi$ appears to be better able of discriminating various judge models, showing more divergence across the tested judges.

To understand how indicative the two alignment metrics are of the expected accuracy of the overall judgement of the models, we plot, for each judge model and exam-taker model, the difference between the score assigned by the judge and the score assigned by a human. In the figure, we can see that for Scott's $\pi$ values higher than 80, the evaluation scores are comparatively close to the human evaluation scores, with a difference of up to 5 points in their assigned scores (complete results table provided in Appendix J). For percent alignment, on the other hand, even judges that have more than 90% may still differ more than 10 points in their assigned score. Interestingly, the deviation from human-judgements for a single judge model can be quite different depending on the exam-taker model. In Figure 1a, `Gemma 2B`, for instance, sometimes assigns higher scores than humans, and sometimes much lower. In the next section, we further explore this particular pattern.

## 4.2 EXPLORING CONSISTENT PATTERNS IN JUDGE MODELS

In the previous section, we have seen that none of the judge models we considered were aligned with humans as well as the humans were aligned amongst themselves. Furthermore, as can be seen in Figure 2, the scores assigned by even the best aligned judge models can differ up to 5 points with the human-assigned scores. However, while this may limit – to some extent – the utility of using a judge models to get a perfect estimate of the exam-taker model's capability on the benchmark, the judge models may still offer valuable insights to *differentiate* between different exam-taker models. If judges exhibit consistent biases such as – akin to a very strict teacher – consistently rating any exam-taker model lower, they will not assign identical scores but may assign identical *rankings*.

To evaluate this, we compare the rankings assigned by each judge model to the nine exam-taker models by computing their Spearman's rank correlation coefficients $\rho$ (Spearman, 1904) with the human ranking. We show the rankings in Figure 3a, with $\rho$ and corresponding $\sigma$ values in Appendix L. Most judge models have rank correlations higher than 0.7; it appears they struggle to distinguish between poorer-performing exam-taker models, but do well at distinguishing between better-performing ones. Notably, the results show that several models that assign scores quite divergent from humans and have poor alignment on the sample level are very aligned in terms of the rankings they assign. Specifically, both `contains` and `Mistral 7B`, with Scott's $\pi$ values of 64 and 66, respectively, exhibit very high rank correlation with the human

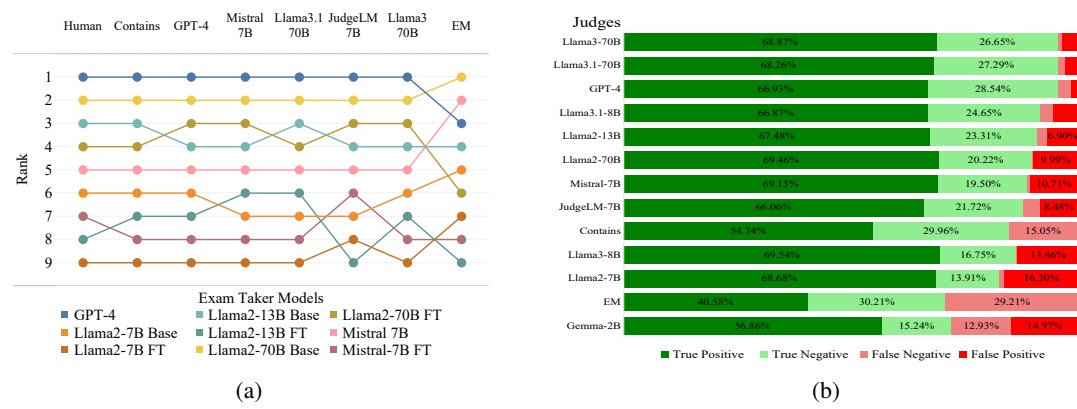

(a)                                                                                          (b)

Figure 3: **Judge rankings and true/false positives and negatives.** (a) Assigned exam-taker model rankings assigned by highly human aligned judges. `Contains` stays closely to human-assigned rankings, as well as `GPT-4 Turbo` and `Mistral 7B`. (b) False positives and negatives across different judge models, in descending order of human alignment. Both false negatives and false positives increase as human alignment decreases, but well-aligned models tend to produce more false negatives than false positives.

scores ($\rho$ 0.99 and 0.98, respectively, with $\sigma$ 0.02 and 0.03). With that, these judges perform on par with `GPT-4 Turbo` and outperform the better `Llama` judges – though with lower significance values – indicating that identifying which models are better should not be equated to assigning them the correct score.

## 5 ANALYSIS

To better understand the judge models, we conduct multiple case studies aimed at identifying common errors and vulnerabilities in the judges we investigate. Specifically, we study their precision and recall and error types (§ 5.1), their sensitivity to the instruction prompt prompt (§ 5.2), how they respond to controlled resposes of specific types (§ 5.3), and the extent to which they have a *leniency bias* (§ 5.4).

### 5.1 BETTER ALIGNED MODELS: PRECISION AND RECALL GAINS WITH ERROR SPOTLIGHTS

We first investigate the precision and recall of the judge models. Maintaining the ordering of Figure 1, we plot both in Figure 4a. We can see that both precision and recall exhibit a moderate increasing trend as alignment increases. In Figure 3b, we observe a similar pattern, though with a clearer picture on the distribution of false positives and negatives. Specifically, we see that the number of true positives is quite stable across many judges. The true negatives, instead, drop off quickly as the judge quality decreases, suggesting it is generally easier to judge answers that are correct.

Next, we analyse the types of errors made by the judge models by manually annotating 900 outputs from Llama-7B Base with error codes, focusing on the top performers `GPT-4 Turbo` and `Llama-3 70B`. We then determine the percentage of each error type that are correctly judged to be incorrect by these two models. The results are shown in Table 2, where it can be observed that both `GPT-4 Turbo` and `Llama-3 70B` have a good error recall when the answers refer to an incorrect entity, or when too many entities are present. Under-specified and otherwise incorrect answers are most challenging for both judges, while answers with too few entities are judged relatively accurately by `GPT-4` but less accurately by `Llama-3 70B`.

### 5.2 JUDGE MODEL SENSITIVITY TO PROMPT LENGTH AND SPECIFICITY

Next, we study the impact of the prompt on the predictions of the judge models, to understand if the success of various judge models is related to the *length* of the prompt, and to study the degree to which

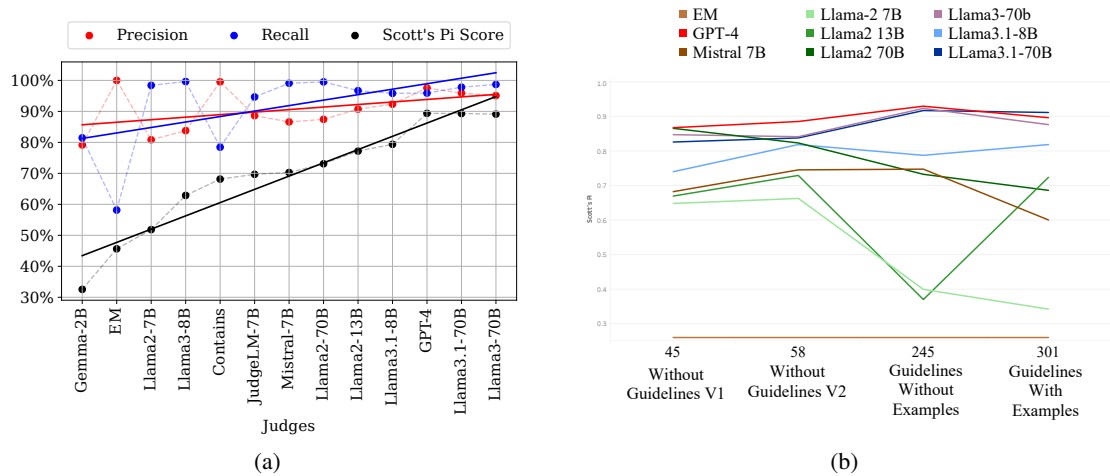

(a)                                                                         (b)

Figure 4: **Precision, recall and prompt sensitivity.** (a) Recall and precision improve with increasing human alignment ($R^2 = 0.31$ and $R^2 = 0.21$, respectively). (b) Scott's $\pi$ scores for judges across different instructions.

Table 2: **Error analysis for `GPT-4` and `Llama-3 70B` judges.** The example question is *"Excluding Lady Jane Grey, who were the five monarchs of the House of Tudor?"*, the correct answer *"Henry VII, Henry VIII, Edward VI, Mary I and Elizabeth I"* (in any order).

| Error code | Explanation | Example | Proportion | GPT-4 recall | Llama-3 70B recall |
|---|---|---|---|---|---|
| **Incorrect entity** | Response refers to a wrong entity | `Henry VII, James I, Edward VI, Mary I and Elizabeth I` | 86.9% | **98.3%** | **96.6%** |
| **Under-specified** | Response contains only part of the answer | `Henry VII, Henry VIII, Edward, Mary, and Elizabeth` | 37.3% | 33.9% | 23.3% |
| **Too few entities** | Response contains too few entities | `Henry VII, Edward VI, Mary I and James I` | 2.47% | **80.0%** | 60.0% |
| **Too many entities** | Response contains too many entities | `Henry VII, Henry VIII, Edward VI, Mary I, James I, and Elizabeth I` | 2.7% | **90.1%** | **90.1%** |
| **Other** | Response is incorrect but cannot be put into any of the above buckets | `I'm sorry but I do not know the answer to that question` | 1.23% | 20.0% | 40.0% |

the judgments of the judge models change with the *specificity* of the prompt. We use four different prompt versions, varying in length and specificity. The first two prompts (`Without guidelines V1/V2`, 45 and 58 tokens, respectively) simply ask to evaluate the responses, without any further information, while more elaborate guidance and examples are given in the longer prompts (`Guidelines without examples` and `Guidelines with examples`, 245 and 301 tokens, respectively). All prompts are listed in Appendix M. Figure 4b shows that `GPT-4 Turbo`, `Llama-3 70B` and `Llama-3.1 70B` exhibit relatively low variance in their agreement with humans as the level of information and the length of the prompt increases. For this task, top performers' (`GPT-4 Turbo`, `Llama-3 70B` and `Llama-3.1 70B`) implicit definition of a correct judgment seems well aligned with the provided instructions and thus shows high alignment with humans even if no specific instructions are provided. It can also be observed that only top performers appears to benefit from the more detailed instructions, with a slight upward trend, whereas the other models get less aligned with more instructions. This might be due to the less powerful judges not being able to follow many instructions in the prompt at the same time. In a follow-up experiment, we further investigate the impact of the order of the reference answers (for details, we refer to Appendix N). Figure 6b illustrates that larger judge models consistently maintain their judgments regardless of the reference order, whereas smaller models – with the exception of `Mistral 7B` – are more sensitive to the reference order given in the prompt.

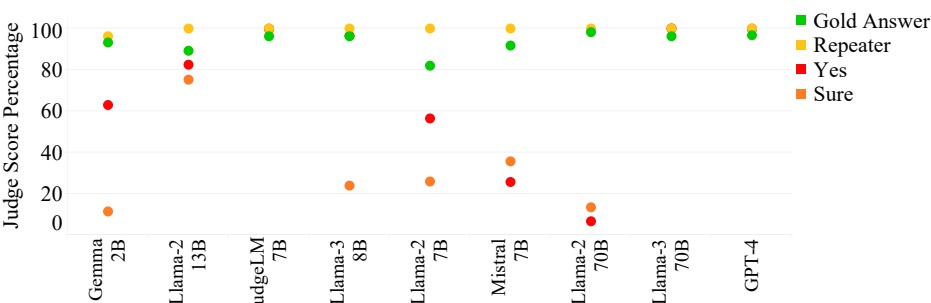

Figure 5: **Judge responses to dummy answers.** We investigate how judge models respond to dummy answers. judge models remain robust when exam-taker models produce responses identical to the prompt ('repeater'), but are less robust when the responses are "Yes" and "Sure". Even when the answer matches one of the reference answers verbatim ('Gold answer'), judges do not always arrive at the correct judgement.

### 5.3 EVALUATING CONTROLLED RESPONSES

Next, we perform simple tests on the judge models by asking them to evaluate a set of dummy benchmark responses. For the first test, the answer to be evaluated for each question is one of the references from the dataset, verbatim (the answer is thus always correct). For the next three tests, the answer is always incorrect. In the second and third tests, the dummy exam-taker model always responds with "Yes", and "Sure" for the second and third tests, respectively. For the fourth test, the evaluated answer is a repetition of the question. In Figure 5, we can see that while some judge models are able to identify and correctly mark the answers as correct (for the first test) or incorrect (for the next three tests), some judges, notably Llama-2 70B, incorrectly evaluate a significant number of dummy answers, even though they show a relatively high alignment with humans on the benchmark evaluations (see Figure 1b). We hypothesise that when the answers are plausible but incorrect (e.g. if the question asks about the name of the author of a book, and the exam-taker model gives the name of the wrong author), most judges are able to identify them as being incorrect (by comparing it with the reference answer). However, the judges might get confused about what they are supposed to evaluate if the answer is completely unrelated to the question (such as the words "Yes" and "Sure"). It is possible that, in this situation, a judge model tries to evaluate one of the reference answers, thus marking it as correct, though further research is required to identify the cause of this behavior.

### 5.4 LENIENCY BIAS IN JUDGE MODELS

Lastly, to get a general sense of the inherent biases or misalignment in the evaluation criteria that might be present in the judge models, we estimate if they have a positive or negative bias in their judgment. To do so, we assume that a judge assigns the correct judgment (i.e. same evaluation as the ground truth) with a probability of $P_c$ and assigns the rest of the samples to be "correct" with a probability $P_+$, which we call their *leniency bias*. We estimate the values of $P_c$ and $P_+$ from the benchmark results[5] and show them in Figure 6a. We observe that $P_+$ for most models is significantly higher than $0.5$, indicating a tendency of the judge models to evaluate responses as "correct" when their evaluation criteria are not completely aligned with the provided instructions.

## 6 CONCLUSION

In this work, we provide an extensive study of the properties of LLMs as judges, comparing them with human judges as well as automated evaluation methods. By focusing on a clean evaluation scenario in

---

[5]The theoretical derivation of the expressions for $P_c$ and $P_+$, as well as the empirical validation for their estimated values can be found in Appendix O.

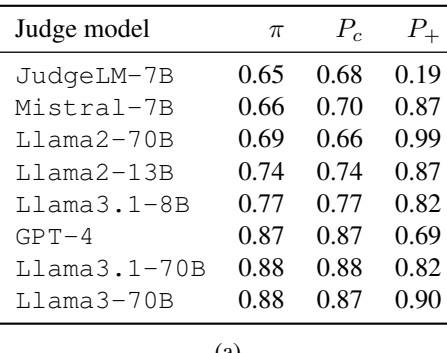

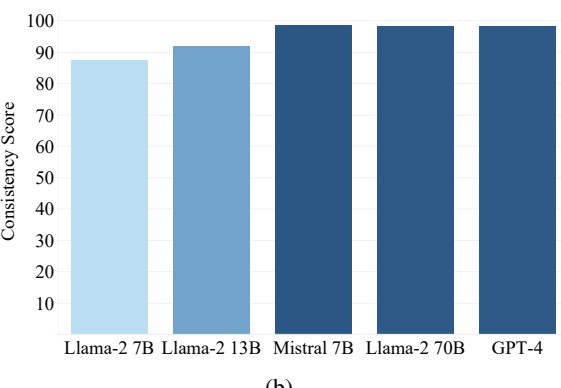

| Judge model | $\pi$ | $P_c$ | $P_+$ |
|---|---|---|---|
| JudgeLM-7B | 0.65 | 0.68 | 0.19 |
| Mistral-7B | 0.66 | 0.70 | 0.87 |
| Llama2-70B | 0.69 | 0.66 | 0.99 |
| Llama2-13B | 0.74 | 0.74 | 0.87 |
| Llama3.1-8B | 0.77 | 0.77 | 0.82 |
| GPT-4 | 0.87 | 0.87 | 0.69 |
| Llama3.1-70B | 0.88 | 0.88 | 0.82 |
| Llama3-70B | 0.88 | 0.87 | 0.90 |

(a)

(b)

Figure 6: **Leniency bias and answer consistency.** (a) Leniency bias across various judge models. (b) Consistency score, defined as the percentage of questions for which the judge model gives the same judgment for three different answer orders.

which inter-human agreement is high, we examine the potential issues with the LLM-as-a-judge paradigm, separately from the ambiguity and subjectivity in the task itself. We find that even in straightforward setups, smaller and more cost-efficient models are less effective judges compared to the best available LLMs, such as Mistral 7B. GPT-4 Turbo, Llama-3.1 70B and Llama-3 70B, instead, are much better aligned, though they are still quite far from the alignment that humans have among each other. In some cases, despite their high alignment, their scores deviate from human scores with up to 5 points. Given the relative simplicity of the scenarios in which we deployed the judges, urging caution in using judge for more complex scenarios. Importantly, we noted that such patterns are virtually undetectable using the commonly deployed metric of *percent aligned*, which barely discrimates between the considered judges. We suggest that future work instead considers the more robust metric Scott's $\pi$, which allows to distinguish judges much better.

Next, we note that to *discriminate* between models, high alignment scores are not an absolute necessity. While GPT-4 Turbo and Llama-3 both have excellent alignment scores, simpler and more cost-efficient and even the lexical matching method contains perform on par when discriminating between the exam-taker models in terms of their *ranking*, despite having much lower alignment scores and score deviations. If the purpose of a study is to determine which model is better and not to estimate their actual scores, such approaches may thus be as suitable as the more expensive ones.

Lastly, we run a range of experiments to investigate judge models' sensitivity to prompts, their precision and recall, their error types, how lenient they are, and how much they can be fooled by dummy answers. We find that LLMs tend to judge positively when in doubt, and this is more pronounced for small models than for larger ones; that judge models with lower alignment lack precision rather than recall, that better models are generally more robust across different prompts, but are difficult to 'steer' in their judgments; that some judge models can be easily fooled by dummy answers such as 'Yes' and 'Sure'; and that judge models are better at detecting completely incorrect answers than partially incorrect ones.

Overall, this work adds to the realm of LLM evaluation research by assessing judges within a clearly defined and objective framework. Our results highlight the utility of using some LLMs as judges but also urge caution in blindly trusting their judgments, even if they are found to be well-aligned with humans. For practitioners using LLMs as judges – regardless of the setup – we recommend not only computing percent agreement, but also Scott's $\pi$, and pairing these with a qualitative analysis to ensure that conclusions from judge models are less susceptible to biases. We further elaborate on the limitations of our work in Appendix A. In the future, we plan to expand our work to increasingly more complex scenarios with more open-ended answers and variability and more generally assess how consistent our findings are across dataset samples, benchmarks, and prompt templates.

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

## A    LIMITATIONS

In our work, we have evaluated how 11 different LLMs fare as judges in a scenario in which judgements should be relatively straight-forward, and human alignment is high. As any study, our work has several limitations as well as directions that we did not explore but would have been interesting too. In this section, we discuss both.

**Simplicity of the task**    As mentioned in the introduction of our work, the scenario in which judges are used are typically much more complicated than the scenario that we focussed on. Specifically, judges are most often deployed in preference rankings (where two model responses are compared) or to judge complex answers that are difficult to automatically parse. In such tasks, human agreement is often low, making it challenging to judge the judges themselves. In our work, we have deliberately chosen for a simple task, in which human alignment is high. The main premise is, that if a judge does not perform well in this simple setup, caution is suggested also in more complex setups – if someone cannot do multiplication, why would they be able to solve ordinary differential equations. Given the poor understanding of which abilities of LLMs generalise in what dimensions, however, more studies are needed to understand how our results generalise to various other scenarios.

**Human alignment**    In an earlier version of this paper, due to the high cost of human annotations, we opted to select a single model for human annotation as we iteratively modified the exam taker prompt, few-shot examples, and guidelines. We selected the Llama2 7B for this purpose with a random sample of 600 questions. As this is only a single model, it is possible that our human alignment scores are biased because of that. After, we have therefore extended our results with another 600 human-annotated examples from Llama3.1 70B. For Llama2 7B The average alignment among human evaluators had a Scott's $\pi$ of $96.36 \pm 1.46$,and the average percent agreement was $98.33\% \pm 0.76\%$. For Llama3.1 70B, we noted that the average alignment among human evaluators had Scott's $\pi$ of $95.78 \pm 0.30$,% and the average percent agreement was $98.72\% \pm 0.10\%$. Given the similarity of these two numbers, we believe that these 1200 samples provide an adequate estimate. In the paper, we take the average.

**Size of the judged samples**    As each of the nine exam-taker models requires human annotations for each sample, we restricted our analysis to 400 samples in total. This sample size also allowed us to conduct manual annotations and error analysis within 75 human hours/200 GPU hours (see Appendix H) and give reliable confidence intervals while also providing the flexibility to compare a range of models. We were not able to increase the size due to the cost, but a statistical analysis (details provided in Appendix I) illustrated that the variance because of this sample size was very low.

**Selection of judges**    With our selection of judges, we have stuck to autoregressive judges that can be used off-the-shelve, as well as one LLM specifically trained to judge. They are – at the moment of writing – the ones that are most commonly used as LLM-judges, and we have tried to be comprehensive across size and family. Nevertheless, we acknowledge that there are other judges that we could have considered as well. As including more judges in – compared to including more exam-taker models– relatively straightforward because it requires only computational power, no manual annotation, we hope that others may evaluate their newly proposed judges using our setup as well.

**Future work**    All in all, these differences underline how finicky using LLMs as judges can be, and with that confirm the overall conclusions of our study that much more work is needed to better understand the strengths and limitations of judge models across a wide range of scenarios and model accuracies. We consider assessing the strengths across multiple different samples and tasks, which would require many more human annotations, outside the scope of this paper and leave such experimentation for future work.

## B   A BRIEF EXPLANATION OF THE THEORETICAL ISSUES WITH COHEN'S KAPPA

Cohen's Kappa Coefficient (Cohen, 1960) is a statistic to measure inter-rater agreement for categorical responses. Cohen's Kappa coefficient measures this agreement by computing the observed (percent) agreement between raters ($p_o$) and comparing it with the hypothetical probability of chance agreement ($p_e$), which is taken as a baseline, as follows:

$$\kappa \equiv \frac{p_o - p_e}{1 - p_e} \tag{1}$$

In this equation, the chance agremeent $p_o$ constitutes the hypothetical probability that observed agreement occurred by chance, given the observed distributions of the considered raters, under the assumption that the probabilities the raters assign to the observed labels are independent. Specifically, it is defined as:

$$p_e = \sum_k \widehat{p_{k12}} =^{ind} \sum_k \widehat{p_{k1}}\widehat{p_{k2}} = \sum_k \frac{n_{k1}}{N}\frac{n_{k2}}{N} = \frac{1}{N^2}\sum_k n_{k1}n_{k2}, \tag{2}$$

where $\widehat{p_{k12}}$ is the estimated probability that rater 1 and rater 2 will classify the same item as $k$, rewritten to $\widehat{p_{k1}}\widehat{p_{k2}}$ under the assumption that $p_{k1}$ and $p_{k2}$ are independent. The crux of the issue with this method of computation, is that $\widehat{p_{k1}}$ and $\widehat{p_{k2}}$ are estimated independently from the data. As such, the chance agreement adjusts for the observed average differences between raters, which is in fact part of what we intend to measure. To address this issue, Scott's Pi (Scott, 1955) instead defines the chance baseline under the assumption that the raters have the same distribution, which is estimated considering the joint distribution of rater 1 and rater 2, rather than considering them separately. It defines $p_e$ as:

$$p_e = \sum_k \widehat{p_k^2} = \sum_k \sum_k (\frac{n_{k1} + n_{k2}}{2N})^2 \tag{3}$$

As such, contrary to Cohen's Kappa, it captures differences surpassing the chance agreement if rater 1 and rater 2 were in fact equivalent. In other words, we compare against a baseline in which raters would be equivalent, and we measure how much they deviate from that.

Note that if the empirical distributions of rater 1 and rater 2 are the same, so will the values of Scott's Pi and Cohen's Kappa be. This also implies that for larger observed (percent) alignment values, the values for Cohen's Kappa and Scott's Pi will be closer.

## C   MODEL AND DATASET DETAILS

In Table 3, we show the different models and datasets used in our experiments, along with version and license details.

## D   MODEL EVALUATION PROMPT TEMPLATES

In Figure 7 and Figure 8, we show the prompt templates used for the base and chat exam-taker models during the question answering process.

## E   JUDGE LLM PROMPT TEMPLATES

In Figure 9, we show the prompt template used to guide the judge models during the evaluation process of a 400-question sample from the TriviaQA unfiltered dataset.

Table 3: Version and license details for the different models and datasets used in experiments.

| Asset | Version | License |
|---|---|---|
| TriviaQA | mandarjoshi/trivia_qa | apache-2.0 |
| Llama-2 7B Base | meta-llama/Llama-2-7b-hf | llama2 |
| Llama-2 7B Chat | meta-llama/Llama-2-7b-chat-hf | llama2 |
| Llama-2 13B Base | meta-llama/Llama-2-13b-hf | llama2 |
| Llama-2 13B Chat | meta-llama/Llama-2-13b-chat-hf | llama2 |
| Llama-2 70B Base | meta-llama/Llama-2-70b-hf | llama2 |
| Llama-2 70B Chat | meta-llama/Llama-2-70b-chat-hf | llama2 |
| Mistral 7B Base | mistralai/Mistral-7B-v0.1 | apache-2.0 |
| Mistral 7B Chat | mistralai/Mistral-7B-Instruct-v0.2 | apache-2.0 |
| Llama-3 8B Chat | meta-llama/Meta-Llama-3-8B-Instruct | llama3 |
| Llama-3 70B Chat | meta-llama/Meta-Llama-3-70B-Instruct | llama3 |
| Llama-3.1 8B Chat | meta-llama/Meta-Llama-3.1-8B-Instruct | llama3.1 |
| Llama-3.1 70B Chat | meta-llama/Meta-Llama-3.1-70B-Instruct | llama3.1 |
| JudgeLM | BAAI/JudgeLM-7B-v1.0 | Non-commercial license |
| GPT-4 Turbo | gpt-4-turbo-2024-04-09 | N/A |

```
Prompt template for Base exam-taker models

Q: Can you name the actress who links 'The Darling Buds of May' and
'Rosemary and Thyme'?
A: Pam Ferris

Q: A neologism is a new?
A: Word/expression

Q: Who, in 2010, became the first person from outside the British
Isles to win the World Snooker Championship title since Cliff Thorburn
in 1980, and the first non British player to win the title since Ken
Doherty in 1997?
A: Neil Robertson

Q: Which German Nazi leader flew solo from Ausberg in 1941 and landed
by parachute near Glasgow on a private peace mission?
A: Hess

Q: Where would you find Narita airport?
A: Tokyo, Japan

Q: Which cartoon title character has a friend called Captain Haddock?
A:
```

Figure 7: Prompt template for base exam-taker models

## F    METRICS FOR JUDGE MODELS

If one of the annotators is taken to be the reference, then the annotations of the other annotator can be categorized as true positives, false positives, true negatives, and false negatives, with the total number of each of them in a benchmark being represented by $T_P, F_P, T_N$, and $F_N$ respectively.

**Prompt template for Chat exam-taker models**

```
You are a part of a question answering benchmark.  Look at the
following examples on how to answer the questions.

'''
Q: Can you name the actress who links 'The Darling Buds of May' and
'Rosemary and Thyme'?
A: Pam Ferris

Q: A neologism is a new?
A: Word/expression

Q: Who, in 2010, became the first person from outside the British
Isles to win the World Snooker Championship title since Cliff Thorburn
in 1980, and the first non British player to win the title since Ken
Doherty in 1997?
A: Neil Robertson

Q: Which German Nazi leader flew solo from Ausberg in 1941 and landed
by parachute near Glasgow on a private peace mission?
A: Hess

Q: Where would you find Narita airport?
A: Tokyo, Japan
'''

Your task is to answer the following question.  Remember to be concise
and only give the answer in a few words.

Q:Which cartoon title character has a friend called Captain Haddock?
A:
```

Figure 8: Prompt template for Chat exam-taker models

```
Prompt template for judge models

Your task is to look at the following question, and based on the
references provided, determine if the model's response is correct or
incorrect.  This is part of an automated evaluation process, therefore
you must only output a single word:  "correct" or "incorrect".

Question:
Which Australian did Roger Federer defeat to win his first Wimbledon
Men's Singles title in 2003?

References:
MARK PHILIPPOUSSIS
MARK PHILIPPOUSSIS

Model Response:
Mark Philippoussis

Evaluation (correct/incorrect):
```

Figure 9: Prompt templates for the judge models

**Percent agreement** is simply the ratio of the numbers of times two annotators agree with each other relative to the total number of annotations. This ratio can have values between $0$ and $1$. For the binary case, the alignment ratio $\rho$ is given as

$$\rho = \frac{T_P + T_N}{T_P + F_P + T_N + F_N}.$$

(4)

**Scott's Pi**, (Scott, 1955), measures the alignment of two annotators while also taking into account the possibility of agreement by pure chance. This coefficient usually has values above $0$ in most real-world situations. The value of Scott's Pi is given as

$$\pi = \frac{p_o - p_e}{1 - p_e} = \frac{\left(\frac{TP+TN}{TP+TN+FP+FN}\right) - \left(\frac{(TP+FP)(TP+FN)+(TN+FN)(TN+FP)}{(TP+TN+FP+FN)^2}\right)}{1 - \left(\frac{(TP+FP)(TP+FN)+(TN+FN)(TN+FP)}{(TP+TN+FP+FN)^2}\right)}$$

(5)

Here, $p_o$ is the relative observed agreement, and $p_e$ is the hypothetical probability of chance agreement.

## G   HUMAN ANNOTATION GUIDELINES

We provide the guidelines used for human evaluation below.

---

**Humn annotation guidelines**

```
You will be given a question, a set of reference answers and the
answer given by an LLM. Your task is to judge if the answer given
by the LLM is correct, as if you were the LLMs teacher grading
their exam.  An answer should be counted as correct if it is
semantically equivalent to (one of the) reference answers.  In
doing so, please follow the following guidelines:
```

- Underspecified answers (e.g.  "December" instead of "December 20") should be marked *incorrect*.
- Answers that have more information than requested (e.g. "December 20, in Paris" instead of "December 20") should be marked correct, provided the extra information is not incorrect or contrasting the rest of the answer.
- Answers with unnecessary verbosity but correct answers should be marked correct (E.g.  "Thanks for asking this question!  The correct answer is:  ...").

```
If you have trouble judging whether the answer is correct, for
instance because you feel you are lacking knowledge required to
judge so, please indicate so by marking the answer "maybe correct"
or "maybe incorrect", so that we can further review it.
```

---

## H   EXPERIMENT COSTS

The costs for the different experiments described in this work belong in three categories – GPU-hours for running open-source models on one or more `Nvidia A100` GPUs, OpenAI credits for making API calls to OpenAI models,[6] and human hours for manual annotations of benchmark responses. The estimated costs for the final reported experiments are given in Table 4. In addition to this, previous unreported experiments and trials had an approximate cost of 120 GPU-hours, 100 USD in OpenAI credits, and 50 human hours, bringing the total experimental cost for this work to approximately 200 GPU-hours, USD 125 OpenAI credits, and 75 human annotation hours.

---

[6]Pricing details for OpenAI models are available at `https://openai.com/api/pricing/`

Table 4: Estimated costs for the final reported experiments. GPU-hours are in equivalent `Nvidia A100` hours, OpenAI credits are in USD, and human hours are time spent in manual annotation.

| Experiment | GPU-hours | OpenAI credits | Human hours |
|---|---|---|---|
| Main benchmarks | 5 | 2 | - |
| Main evaluations | 30 | 8 | 10 |
| Human alignment | 2 | - | 9 |
| Error analysis | 1.5 | - | 5 |
| Controlled responses | 15 | - | - |
| Leniency bias | 5 | 5 | - |
| Guideline bias | 10 | 5 | 1 |
| Reference bias | 5 | 4 | 1 |
| **Total** | **73.5** | **24** | **26** |

## I   STATISTICAL RELIABILITY OF EVALUATION SAMPLE

Due to computational constraints discussed in Appendix A and Appendix H, we limit our evaluation set to randomly sampled 400 questions from TriviaQA (Joshi et al., 2017). In this section, we further take 5 samples of 300 randomly selected questions from the evaluation set and calculate the mean and standard deviation of Scott's Pi. From Table 5, it can be observed that even on down-sampled sets, the Scott's $\pi$ values are similar to Figure 1b. Standard deviation of all the judge models from the mean Scott's $\pi$ is also minimal, barring `EM` lexical match.

Table 5: Weak Scott's $\pi$ variation for the 5 down-sampled sets indicating robustness for the evaluation sample

| Judge Model | Mean Scott's $\pi$ | Std Dev |
|---|---|---|
| Llama3-70B | 0.88 | 0.0046 |
| Llama3.1-70B | 0.88 | 0.0039 |
| Llama3.1-8B | 0.78 | 0.0050 |
| Llama2-13B | 0.75 | 0.0043 |
| Llama2-70B | 0.69 | 0.0114 |
| Mistral-7B | 0.67 | 0.0108 |
| JudgeLM-7B | 0.66 | 0.0026 |
| Contains | 0.64 | 0.0087 |
| Llama3-8B | 0.60 | 0.0126 |
| Llama2-7B | 0.47 | 0.0112 |
| EM | 0.47 | 0.29 |
| Gemma-2B | 0.26 | 0.007 |

## J   JUDGE SCORES

We show the scores assigned by each judge model to each exam-taker model, visualised in Figure 1a in Table 6.

## K   EXAM-TAKER MODEL BASE VS CHAT ANALYSIS

Given the human judgments we have available, we take the opportunity to investigate the performance differences between base and their corresponding chat models. In Table 7, we show the scores assigned by

Table 6: Judge model score card for every exam-taker model.

| | **Exam taker models** | | | | | | | | |
| | Llama2 | | | | | | Mistral | | GPT-4 |
| | | Base | | | Chat | | Base | Instruct | |
| **Judge Models** | 7B | 13B | 70B | 7B | 13B | 70B | | 7B | |
| Llama 3.1 8B | 65.25 | 75.00 | 83.50 | 60.25 | 70.50 | 75.50 | 73.75 | 59.00 | **89.00** |
| Llama 3.1 70B | 62.00 | 74.25 | 85.00 | 55.50 | 64.75 | 74.00 | 72.25 | 60.50 | **92.25** |
| Llama 3 8B | 76.00 | 83.25 | 91.50 | 73.25 | 82.75 | 85.25 | 81.75 | 76.0 | **97.25** |
| Llama 3 70B | 64.25 | 75.50 | 86.50 | 57.00 | 64.00 | 75.75 | 73.5 | 62.50 | **92.75** |
| Llama 2 7B | 80.50 | 85.25 | 92.00 | 80.50 | 70.75 | 90.75 | 84.00 | 83.25 | **97.75** |
| Llama 2 13B | 68.25 | 75.50 | 86.50 | 63.25 | 62.75 | 77.50 | 74.50 | 67.50 | **93.5** |
| Llama 2 70B | 71.25 | 80.5 | 90.25 | 67.50 | 74.75 | 81.25 | 80.0 | 72.5 | **96.75** |
| Mistral 7B | 72.50 | 80.75 | 90.50 | 69.00 | 74.75 | 82.50 | 80.25 | 72.00 | **96.25** |
| Gemma 2B | 79.75 | 87.00 | **91.25** | 58.50 | 41 | 68.50 | 84.0 | 55.75 | 80.50 |
| JudgeLM | 69.50 | 77.75 | 86.25 | 63.75 | 48.0 | 82.75 | 77.25 | 71.0 | **94.50** |
| GPT-4 | 60.50 | 71.50 | 82.50 | 54.50 | 59.0 | 73.0 | 69.75 | 56.50 | **90.0** |
| Exact Match | 46.75 | 56.00 | **63.75** | 24.00 | 0.25 | 36.25 | 59.50 | 20.25 | 58.25 |
| Contains Match | 50.75 | 60.00 | 68.00 | 39.00 | 46.25 | 59.50 | 57.25 | 44.00 | **70.00** |
| Human Eval | 62.50 | 72.75 | 83.75 | 56.00 | 56.50 | 72.25 | 71.75 | 60.75 | **91.50** |

various judge models to four base-chat pairs. According to the default metric EM, the base models outperform the chat models by a large margin. Interestingly, while this difference gets smaller when the answers are judged by humans (second column) or GPT-4 Turbo, there is still a substantial difference for all four pairs, suggesting that the difference is not merely an effect of the increased verbosity of the chat models. Further evidence for that hypothesis is provided by Figure 10b, in which we can see that while 14% of the errors are shared between the base-chat pairs, almost another 14% of the examples get judged correctly by the base models but not by the chat models, while the opposite happens in only 2.5% of the cases. We consider two alternative hypotheses:

i) The chat models have a worse understanding of the particular prompt format, which is tuned more to fit base models; or

ii) The chat models have 'unlearned' some knowledge during their alignment training.

Table 7: Scores of base and chat models by various judges

| Base-Chat pair | **Judge models** | | | | | | | | | |
| | EM | | Contains | | Human | | GPT-4 Turbo | | Llama-3 70B | |
| | Base | Chat | Base | Chat | Base | Chat | Base | Chat | Base | Chat |
| Llama-2 7B | **46.75** | 24.00 | **50.75** | 39.00 | **62.25** | 56.00 | **60.50** | 54.50 | **64.25** | 57.00 |
| Mistral 7B | **59.50** | 20.25 | **57.25** | 44.00 | **71.75** | 60.75 | **69.75** | 56.50 | **73.50** | 62.50 |
| Llama-2 13B | **56.00** | 0.25 | **60.00** | 46.25 | **72.75** | 56.50 | **75.00** | 59.00 | **76.50** | 64.00 |
| Llama-2 70B | **63.75** | 36.25 | **68.00** | 59.50 | **83.75** | 72.25 | **82.50** | 73.00 | **86.50** | 75.75 |

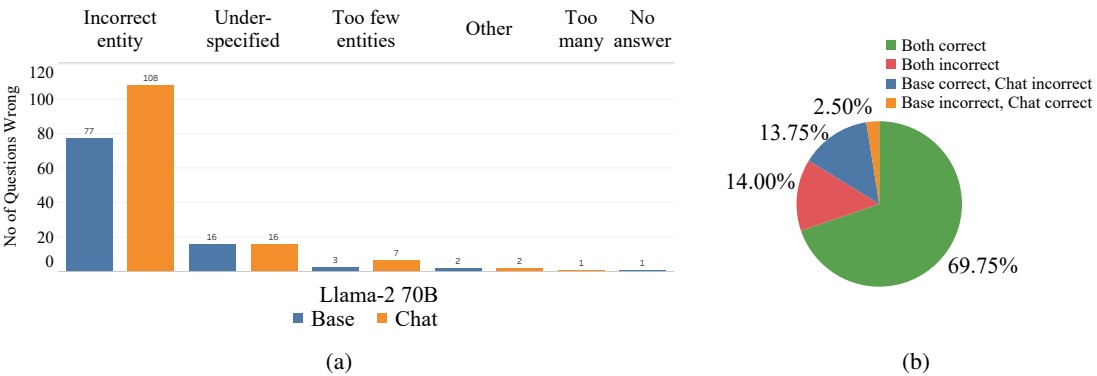

Figure 10: a) Distribution of incorrect question counts by error codes for the Llama2 70B Base vs Chat exam-taker models evaluated on 400 questions. b) Pie chart showing the percentage of questions categorized by the judgment from Base and Chat models.

To disentangle these two factors, we manually analyse 400 questions for Llama-2 70B and Llama-2 70B-chat, using our earlier error codes. The results, shown in Figure 10a, suggest that, at least to some extent, the difference between base and chat models is in fact due to 'unlearning' of knowledge: while the number of errors is more or less equal among most categories, there is a stark difference in the incorrect entity category. Substantially more often than the base models, the chat models do answer the question with a semantically plausible but incorrect entity. In Table 8-Table 10, we provide examples of such cases. The results do not show any evidence to support the first hypothesis: the number of errors where the answer cannot be parsed or is just entirely incorrect does not differ between base and chat models.

Table 8: Knowledge unlearning example 1.

| Question: Which British artist's works include 'The First Real Target'? | |
|---|---|
| **References** | Peter Blake, Peter Balke, Sir Peter Blake |
| **LLama-2 70B Base** | Peter Blake |
| **LLama-2 70B Chat** | Patrick Caulfield |
| **Mistral 7B Base** | David Hockney |
| **Mistral 7B Chat** | Damien Hirst |

Table 9: Knowledge unlearning example 2

| Question: Who was the first cricketer to score 10,000 test runs? | |
|---|---|
| **References** | Sunil Gavaskar, Sunil Manohar Gavaskar, SM Gavaskar, Sunny gavaskar, Gavaskar |
| **LLama-2 70B Base** | Sunil Gavaskar |
| **LLama-2 70B Chat** | Sachin Tendulkar |
| **Mistral 7B Base** | Sachin Tendulkar |
| **Mistral 7B Chat** | Sachin Tendulkar was the first cricketer to score 10,000 runs in Test matches. |

Table 10: Knowledge unlearning example 3

| Question: 'Uncle Harry's Coat' was the first garment produced by which famous jacket manufacturer, based in Simonside, Newcastle Upon Tyne? | |
|---|---|
| **References** | Barbour |
| **LLama-2 70B Base** | Barbour |
| **LLama-2 70B Chat** | Barbour |
| **Mistral 7B Base** | Barbour |
| **Mistral 7B Chat** | Jack Walker & Sons |

## L    EXAM-TAKER MODEL RANKING CORRELATION

In Appendix L, We use the Spearman Rank correlation coefficient (Spearman, 1904) to assess the rankings of the exam-taker models. To validate these rankings, we randomly select 6 out of 9 exam-taker models across 5 samples, subsequently calculating the mean ($\rho$) and standard deviation ($\sigma$) of the rankings. The results reveal that the `contains` model exhibits the highest stability and $\rho$ among the rankings, while the majority of judge models achieve a coefficient exceeding 0.7, indicating a strong alignment. Notably, smaller models such as `Mistral 7B` perform on par with `GPT-4 Turbo`, highlighting the robustness of smaller models in maintaining rankings.

Table 11: Spearman Rank Correlation Coefficient $\rho$.

| Judges | $\rho$ | $\sigma$ |
|---|---|---|
| Contains | 0.99 | 0.02 |
| Mistral-7B | 0.98 | 0.03 |
| GPT-4 | 0.98 | 0.03 |
| Llama2-13B | 0.95 | 0.18 |
| JudgeLM-7B | 0.95 | 0.05 |
| Llama2-7B | 0.94 | 0.04 |
| Llama3.1-70B | 0.94 | 0.07 |
| Llama3-70B | 0.93 | 0.05 |
| Llama3.1-8B | 0.89 | 0.10 |
| Llama3-8B | 0.86 | 0.07 |
| Llama2-70B | 0.84 | 0.13 |
| Gemma-2B | 0.71 | 0.20 |
| EM | 0.67 | 0.13 |

## M    TOO MUCH INFO CONFUSES JUDGES

In Figure 11-14, we report the guidelines we used for the experiments in § 5.2. The simplest prompt used is *Without Guidelines v1* (see Figure 11) where we define a sequential and structured process for the judge model. In *Without Guidelines v2* (see Figure 12), we add an additional focus on the overall task and outcome as well. For *Guidelines without examples* (see Figure 13), we provide the judge models with detailed instructions about the task at hand, along with explicit guidelines on how to evaluate the answers. Additionally, for *Guidelines with examples*(see Figure 14), we also provide examples to the judge models for further reference.

```
Without Guidelines v2 (58 tokens)

Your task is to look at the following question, and based on the
references provided, determine if the model's response is correct or
incorrect.  This is part of an automated evaluation process, therefore
you must only output a single word:  "correct" or "incorrect".

Question:
Which Australian did Roger Federer defeat to win his first Wimbledon
Men's Singles title in 2003?

References:
MARK PHILIPPOUSSIS
MARK PHILIPPOUSSIS

Model Response:
Mark Philippoussis

Evaluation (correct/incorrect):
```

Figure 12: *Without Guidelines v2* prompt template for the judge models

```
Without Guidelines v1 (44 tokens)

Review the question and examine the references provided, then evaluate
the model's response.  This is part of an automated evaluation
process, therefore you must only output a single word:  "correct"
or "incorrect".

Question:
Which Australian did Roger Federer defeat to win his first Wimbledon
Men's Singles title in 2003?

References:
MARK PHILIPPOUSSIS
MARK PHILIPPOUSSIS

Model Response:
Mark Philippoussis

Evaluation (correct/incorrect):
```

Figure 11: *Without Guidelines v1* prompt template for the judge models

**With Guidelines but no examples (245 tokens)**

```
Review the Question and Examine the Reference:  Carefully read
and understand the question.  Carefully read each of the reference
provided for the question:
Evaluate the LLM Response based on guidelines (1-4):
1.  Given the model response, only consider the part(s) of the
response answering the question, and ignore any extra text.
2.  When evaluating the language model's answer, evaluate True if it
matches with at least one of the provided references otherwise False.
3.  An answer and reference pair is considered a match if they are
semantically equivalent given the context of the question.  This
determination should not require world knowledge from the human, and
if it does, consider it not a match.  The answer must contain all the
information of the reference for it to be considered a match.
4.  Considering there might be multiple correct answers, each answer
must match with at least one reference for the response to be
considered correct overall.
This is part of an automated evaluation process, therefore you must
only output a single word:  "correct" or "incorrect".

Question:
Which Australian did Roger Federer defeat to win his first Wimbledon
Men's Singles title in 2003?

References:
MARK PHILIPPOUSSIS
MARK PHILIPPOUSSIS

Model Response:
Mark Philippoussis

Evaluation (correct/incorrect):
```

Figure 13: *Guidelines without examples* Prompt template for the judge models

**With Guidelines and examples (301 tokens)**

```
You will be given a question, a set of reference answers and the
answer given by an LLM. Your task is to judge if the answer given
by the LLM is correct, as if you were the LLMs teacher grading their
exam.   An answer should be counted as correct if it is semantically
equivalent to (one of the) reference answers.  In doing so, please
follow the following guidelines (1-3):
1.   Underspecified answers (e.g.  "December" instead of "December 20")
should be marked incorrect
2.   Answers that have more information than requested (e.g.  "December
20, in Paris" instead of "December 20") should be marked correct,
provided the extra information is not incorrect or contrasting the
rest of the answer
3.   Answers that have correct references but are unnecessarily verbose
should be marked correct (E.g.  "Thanks for asking this question!  The
correct answer is:  ...)
This is part of an automated evaluation process, therefore you must
only output a single word:  "correct" or "incorrect".
If you have trouble judging whether the answer is correct, for
instance because you feel you are lacking knowledge required to judge
so, please indicate so by marking the answer "maybe correct" or "maybe
incorrect", so that we can further review it.

Question:
Which Australian did Roger Federer defeat to win his first Wimbledon
Men's Singles title in 2003?

References:
MARK PHILIPPOUSSIS
MARK PHILIPPOUSSIS

Model Response:
Mark Philippoussis

Evaluation (correct/incorrect):
```

Figure 14: *Guidelines with Examples* Prompt template for the judge models

## N  JUDGE MODELS ARE SENSITIVE TO REFERENCE ORDER

We investigate the judges' sensitivity to reference order by providing the same prompt, question and model response to the judge models, but shuffling the reference order in three different permutations. We compute the consistency score of the model as the percentage of questions for which it gives the same judgment all the 3 times. We observe that the model is more likely to evaluate an answer as correct if the corresponding reference appears early in the list of references (see Figure 15). The smaller judge models sometimes fail to capture all the information in the prompt, and provide judgement based on their own knowledge rather than going by the references (see Figure 16).

Figure 15: Example of `Llama2-7B` getting confused when the order of the references are changed

Figure 16: Example of `Llama2-7B` failing to identify the task by changing the order of the references.

## O  LENIENCY BIAS

As described in § 5.4, for the purpose of the leniency bias experiments, we assume that a judge assigns the correct judgment with a probability of $P_c$ and randomly assigns the rest of the samples to be "correct" with a probability $P_+$. In this section, we derive the mathematical expressions for $P_c$ and $P_+$. We assume that in the case of misalignment between the evaluation criteria of guidelines and judge models, the probability of getting an evaluation of "correct" is independent of the actual correctness of the answer (i.e. the judge model effectively flips a coin to give out its judgement). For any given benchmark and judge model, we denote the ground-truth score as $s$, and the true positive and true negative rates as $t_P$ and $t_N$, respectively, all normalized to be between $0$ and $1$.

Now, based on our assumptions, the true positives, where the exam-taker model response is correct, and also correctly identified by the judge model to be correct, would be comprised of two possible cases: 1) The judge evaluates it correctly according to the given evaluation criteria with a probability of $P_c$; and 2) The judge does not evaluate it according to the given criteria with a probability of $1 - P_c$, but the evaluation still happens to be correct with a probability of $P_+$. With the total ratio of the correct responses being $s$, the true positive rate is therefore given by –

$$t_P = s[P_c + (1 - P_c)P_+] \tag{6}$$

Similarly, the true negatives, where the exam-taker model response is incorrect, and also correctly identified by the judge model to be incorrect, would also be comprised of two cases: **1)** The judge evaluates it correctly according to the given evaluation criteria with a probability of $P_c$.**2)** The judge does not evaluate it according to the given criteria with a probability of $1 - P_c$, but the evaluation still happens to be correct with a probability of $1 - P_+$. With the total ratio of the incorrect responses being $1 - s$, the true negative rate is therefore given by –

$$t_N = (1 - s)[P_c + (1 - P_c)(1 - P_+)]. \tag{7}$$

Using Equation (7), we can derive the following:

$$t_N = (1 - s)[P_c + (1 - P_c)(1 - P_+)] \tag{8}$$
$$= P_c + 1 - P_+ - P_c + P_c P_+ - sP_c - s + sP_+ + sP_c - sP_c P_+ \tag{9}$$
$$= 1 - P_+ + P_c P_+ - s + sP_+ - sP_c P_+ \tag{10}$$
$$= 1 - s - P_+(1 - P_c - s + sP_c) \tag{11}$$
$$= 1 - s - P_+(1 - s)(1 - P_c) \tag{12}$$
$$\implies P_+ = \frac{1 - s - t_N}{(1 - s)(1 - P_c)} \tag{13}$$
$$= \frac{1 - \frac{t_N}{1-s}}{1 - P_c} \tag{14}$$

Substituting the value of $P_+$ in Equation (6), we get:

$$t_P = s[P_c + (1 - P_c)P_+] \tag{15}$$

$$= s\left[P_c + (1 - P_c)\frac{1 - \frac{t_N}{1-s}}{1 - P_c}\right] \tag{16}$$

$$= s\left[P_c + 1 - \frac{t_N}{1-s}\right] \tag{17}$$

$$\implies \frac{t_P}{s} = P_c + 1 - \frac{t_N}{1-s} \tag{18}$$

$$\implies P_c = \frac{t_P}{s} + \frac{t_N}{1-s} - 1 \tag{19}$$

The values of $P_c$ and $P_+$ can be estimated from observed data using the derived expressions. The estimated probabilities using this method, with human evaluation as the reference, are shown in Figure 17a.

To validate these derived values, we observe the correlation between the estimated values of $P_c$ and Scott's Pi ($\pi$). As shown in Figure 17b, we observe that the estimated values of $P_c$ are highly correlated to the Scott's $\pi$ values for the judge models, with a Pearson correlation coefficient of $0.98$.

| Judge model | $\pi$ | $P_c$ | $P_+$ |
|---|---|---|---|
| Gemma-2B | 0.26 | 0.38 | 0.87 |
| Llama2-7B | 0.47 | 0.63 | 0.75 |
| Llama3-8B | 0.59 | 0.63 | 0.74 |
| JudgeLM-7B | 0.65 | 0.68 | 0.19 |
| Mistral-7B | 0.66 | 0.70 | 0.87 |
| Llama2-70B | 0.69 | 0.66 | 0.99 |
| Llama2-13B | 0.74 | 0.74 | 0.87 |
| Llama3.1-8B | 0.77 | 0.77 | 0.82 |
| GPT-4 | 0.87 | 0.87 | 0.69 |
| Llama3.1-70B | 0.88 | 0.88 | 0.82 |
| Llama3-70B | 0.88 | 0.87 | 0.90 |

(a)

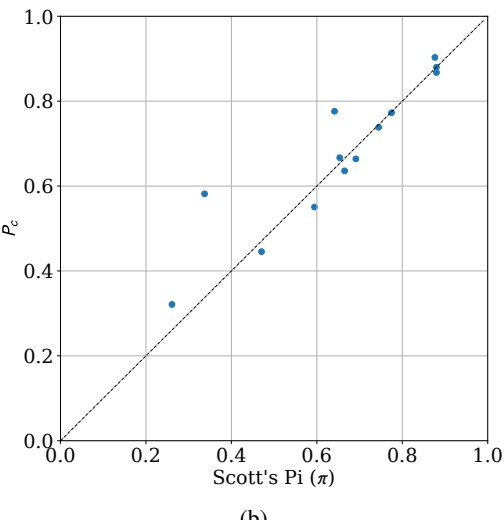

(b)

Figure 17: a) Estimated values of $P_c$ and $P_+$ for different judge models. b) Pearson's correlation coefficient between $\pi$ and $P_c$ for judge models.

