# OpenReview forum: "Judging the Judges: Evaluating Alignment and Vulnerabilities in LLMs-as-Judges"
_ICLR.cc/2025/Conference — ICLR 2025 Conference Withdrawn Submission_

### Official Review · Reviewer_ZKfb · 2024-10-22

**Soundness:** 4
**Presentation:** 3
**Contribution:** 3
**Rating:** 8
**Confidence:** 4

**Summary:**

This paper presents a study on the performance of LLM-as-a-judge expressed as agreement with human assessment. The LLMs that are judged are run in an exam-taker context, using TriviaQA dataset. The study shows that only the largest and more recent Lllama-3.1 models approach human judgement. The rest observe widely different scores, some as low as 60% agreement. Different prompting styles seem to make a difference, with single digit performance improvements for the performant models.

I think it would be nice to summarize all the takeaways in a section and best practices for doing assessment using LLM-as-a-judge.

**Strengths:**

* Thorough and timely study
* Several interesting experiments

**Weaknesses:**

* I would have liked to see more datasets; in fact, I would suggest reducing the number of exam-taker (e.g., I find the base models less interesting) and use different datasets

**Questions:**

Do you think the results generalize to different types of datasets/tasks?

What do you think it's an acceptable agreement percentage?

What are the takeways from this study? What are the best practices that can be adopted when using LLM-as-a-judge?

---

> ### Author Response · Authors · 2024-11-26
>
> We would like to thank the reviewer for their encouraging evaluation of the paper. Below, we respond to the reviewer’s queries raised on paper weaknesses/questions. For context, we first talk about key takeaways.
>
> **1. Some key takeaways -**
> * Advise against uncritical reliance on Judge LLM performance metrics, noting that even in highly controlled environments, state-of-the-art models still lag at least 9% behind human alignment.
> * Recommend using smaller models also such as Llama 3.1 8B and Mistral 7B for objective binary evaluation and ranking tasks, respectively.
> * Advocate for prompt engineering to optimize human alignment first and then use it on Judge LLMs, while cautioning against excessive instruction complexity.
>
> **2. Future:** We’re planning to include more datasets like LLMBar that have an objective Q&A setup.
>
> **3. Generalizability:** These findings likely extend to diverse datasets and tasks. If the scott’s agreement is 88% even in this controlled setup, then a more complex setup suggests potentially greater challenges in more complex scenarios, such as open-ended questions with or without gold standards.
>
> **4. Human alignment thresholds vary across datasets:** Previous literature has noted ~80-85% human alignment for open-ended tasks [1][2] and ~90-96% for objective evaluations [3]. Our work achieves a 98% agreement percentage across 600 questions. During experimentation, we noticed the tradeoff to accepting lower agreement percentage is that it limits the potential optimization of Judge LLMs due to inherent dataset ambiguity. So, Judge LLM performance reaches its peak when the dataset is least ambiguous, and their limitations are inherently tied to the level of uncertainty present in the data they are evaluating.
>
> **References -**
>
> [1] Judging LLM-as-a-Judge with MT-Bench and Chatbot Arena (Zheng et al., NeurIPS 2023)
>
> [2] Strong and weak alignment of large language models with human values (Kharmassi et al., Nature 2024)
>
> [3] Evaluating Large Language Models at Evaluating Instruction Following (Zeng et al., ICLR 2024)

---

### Official Review · Reviewer_oNkj · 2024-10-30

**Soundness:** 1
**Presentation:** 2
**Contribution:** 1
**Rating:** 5
**Confidence:** 4

**Summary:**

The authors examine models as judges on TriviaQA questions, comparing how often humans agree with their answers. While they find alignment is high, they can use Scott Pi metrics to distinguish the quality of the judges. They also find models are frequently lenient judges in a binary setting of “correct” vs “incorrect”.

**Strengths:**

* The paper is clear and nicely visualizes the relevant findings
* The authors explore a dozen models as judges
* The authors use manual annotation to carefully unpack the judge behavior, especially the observation about judge leniency

**Weaknesses:**

* As a reader I'm having difficulties understanding the overarching goals of the paper. Typically researchers use LLMs as a judge for longer form, more subjective questions, where answer coherence, style, and correctness are all part of the judgement. But for TriviaQA, the chosen dataset, the questions have clear, short answers with reference documents, meaning Exact Match is already a strong metric. Here, humans are simply reporting the binary value “correct” vs “incorrect” on the model answers, which seems to have little to do with “human alignment” and more to do with which model got the right answer? Could the authors provide more information on how they think these insights may or may not generalize to more more complex judging tasks, as well as discuss the limitations of their findings?
* The choice of TriviaQA is extremely relevant to the reported results. Could the authors justify this choice? And have they considered comparing their results on other types of datasets?
“well-aligned models tend to produce more false negatives than false positives.” This does not seem to be supported by Figure 3b, where the most correct model’s errors are mostly false positive? Could the authors please provide details to explain this—in case I am misunderstanding?
* Could the authors add more details discussing the extent to which their contributions are novel and provide the community with actionable? A discussion section would be helpful here.

**Questions:**

* Did the authors consider other datasets, or non-binary notions of answer quality?
* Did the authors consider evaluating the alignment across models to understand how they might be ensembled to mimic a better judge?

---

> ### Author Response · Authors · 2024-11-26
>
> We appreciate the reviewer's assessment. To address the identified shortcomings, we have prepared a point-by-point response to each of the concerns mentioned. To add context, contributions to the community is talked first.
>
> **1. Contributions to the community:** These are some of the contributions that we’d like to highlight that has a direct impact on the research community and industry -
>
> * The fact that the judges and humans differ in agreement in a comparatively simpler setup suggests that they might struggle in more complex setups such as open-ended questions with or without gold references. The limitations of our work are discussed in-depth in Appendix A.
> * Large Judge LLMs show strong alignment with humans, but smaller models like Llama 3.1 8B and Mistral 7B also demonstrate impressive performance in ranking and accuracy.
> * Our research quantitatively reveals Judge LLMs' tendency towards leniency (Section 5.4). We also found that excessive instructions can confuse LLMs (Section 5.2), resulting in lenient responses during binary or spectrum evaluations.
>
> **2. Goal of the paper and choice of TriviaQA:** We sought to establish an upper bound on Judge LLM performance in ideal conditions. This approach provides a clear benchmark for the community. We created an unambiguous, "sterile" setup to isolate core capabilities and assess maximum potential without confounding variables. In our preliminary study, when we started with Natural Questions, we couldn’t reach human alignment of ~90% because of ambiguous questions such as “Where did the term liberal arts come from ?”. The MT-Bench paper [1], which also concluded GPT-4 as leading Judge LLM, also observed that human alignment was ~81%. In this scenario, it’s extremely hard to optimize Judge LLMs when the dataset itself is so ambiguous that humans can’t agree to a correct answer.
>
> To address this, we chose TriviaQA dataset which contains factual questions with multiple references which eliminates the possibility of error in judgment being because of ambiguity in the questions/references. Consequently, we achieved 98% percentage agreement and 96% average alignment (Scott’s Pi) on a sample size of 600 between 3 annotators, demonstrating the reliability of our evaluation setup even further (discussed more in Appendix I).
>
> **3. False positive rate of Judge LLMs:** Thank you for bringing this to our attention! We’ll fix the text in paper to reflect that well aligned models tend to produce more false positives than negatives. This is further backed by Section 5.4 where we discuss that Judge LLMs tend to be more lenient.
>
> Having read the rebuttal, would you be willing to adjust your evaluation score? Thank you!
>
> **References -**
>
> [1] Judging LLM-as-a-Judge with MT-Bench and Chatbot Arena (Zheng et al., NeurIPS 2023)

---

> > ### Comment · Reviewer_oNkj · 2024-11-29
> >
> > I would like to thank the authors for their clarifications and answers. Their contributions seem clearer now, and as such I will increase the score from 3 to 5. There are still open concerns with the generalizability of the findings from TriviaQA and this setup, which limits how informative the paper might be. Including the results on Natural Questions would also be of interest.

---

### Official Review · Reviewer_MJzj · 2024-11-04

**Soundness:** 3
**Presentation:** 3
**Contribution:** 3
**Rating:** 5
**Confidence:** 4

**Summary:**

This paper evaluates the LLM-as-a-judge paradigm by comparing 13 judge models against human evaluations on TriviaQA, focusing on a scenario with high human agreement to isolate judge performance issues.

The results show that while the largest models achieve reasonable human alignment, they still fall notably short of human-human agreement levels and exhibit various vulnerabilities.

**Strengths:**

* The explanation of why Scott's Pi should be used instead of Kappa in judging-the-judges scenarios is a significant contribution that will benefit future researchers.
* The comprehensive analysis across multiple dimensions (alignment metrics, ranking correlation, error analysis) provides valuable insights into the strengths and limitations of different judge models.
* The comparison between LLM judges and lexical judges (EM, Contains) offers a novel and important perspective. This insight becomes increasingly critical as NLP tasks grow more complex, helping inform efficient evaluation strategies.

**Weaknesses:**

* The evaluation relies solely on TriviaQA, making it difficult to deconfound the root cause: whether the best model's performance stems from better alignment, knowledge of TriviaQA content, or simply being favored by other LLMs. Other unusual findings may also be specific to TriviaQA: in Figure 1.a, EM's instability compared to Contains likely results from references providing multiple correct answers.
* The paper lacks sufficient content for an ICLR long paper. I suggest expanding the scope by:
    * Including more evaluation datasets covering other types of tasks, such as objective long answers (e.g., code writing), using LLM judges to rank exam takers, etc.
    * Moving Appendix B (issues with Kappa) to the main paper and adding more experiments and analysis. This lesser-known fact would make the paper more informative and impactful.

**Questions:**

see weakness

---

> ### Author Response · Authors · 2024-11-26
>
> We are grateful for the reviewer's evaluation. In response to the concerns raised about weaknesses, we have provided an itemized reply addressing each point.
>
> **1. Evaluation relies solely on TriviaQA:** We intentionally chose TriviaQA for our study, as it provides a well-defined benchmark to assess LLM judging capabilities while minimizing potential ambiguities in the evaluation process. TriviaQA's multiple references per question provide a more robust ground truth, and our inter-annotator agreement (98% on 600 instances) surpasses other datasets like LLMBar's (94% on 80 instances). Our larger sample size offers greater statistical power and sets a clear upper bound for human alignment in factual Q&A tasks. While additional datasets may be valuable for future work, our current methodology provides a focused, reliable assessment of LLM judging capability, supported by high inter-annotator agreement and a substantial sample size.
>
> **2. Lack of scope:**
> * Integrating code writing and evaluation would be an excellent extension of this work. However, smaller models currently struggle to generate high quality code comparable to larger SOTA models. Expanding our infrastructure for code execution requires additional funding and additional work to explore future ambiguities and iterate on new human guidelines. Our current setup is valuable as it accurately simulates how Judge LLMs are used in industry, providing relevant insights into their real-world performance.
>
> * We’ll be happy to move issues with Kappa in the main paper as we also believe that this lesser known fact will help practitioners adopt Scott’s pi metric over Cohen’s kappa.
>
> Having read the rebuttal, would you be willing to adjust your evaluation score? Thank you!

---

### Official Review · Reviewer_pE8n · 2024-11-05

**Soundness:** 2
**Presentation:** 2
**Contribution:** 2
**Rating:** 3
**Confidence:** 3

**Summary:**

The paper offers a study of LLMs-as-judges. The authors investigate 13 models (2B to 70B), evaluating 9 different "exam-taker" models on the TriviaQA benchmark. They found 1) Only the largest models achieve reasonable alignment with humans, though still falling short of inter-human agreement, 2) Scott's π provides better discrimination between judges than percent agreement, and 3) Even models with lower alignment scores can effectively rank exam-taker models. Through detailed analysis, the paper uncovers several vulnerabilities in judge models, including sensitivity to prompt complexity and a tendency toward leniency.

**Strengths:**

- They focus on a specific scenario with high inter-human agreement which is an attempt to isolate the judge model behavior from task ambiguity.
- Several dimensions are explored: 1) model sizes and families, 2) Multiple metrics, 3) Error analysis provided.
- Insights such as ``smaller models can rank exam-takers as effectively as larger ones'', and the attempted explanation that "chat models may "unlearn" some knowledge during alignment";
- The work also provides some recommendations for practitioners using LLM as judges, e.g. using Scott's $\pi$ along with accuracy.

**Weaknesses:**

- The scope remains limited to TriviaQA. For "short, factual" answers, consider adding the "LLMBar" datasets, which have high human agreement rates > 90%. Sufficient examples can be used according to your dataset selection criteria [1]. Without the inclusion of additional datasets [1], it remains unclear how well the ranking ability would transfer.
- The original claim (line 316-318) about judge performance being worse at identifying correct answers could be an artifact of including metrics that are overly strict about exact wording matches rather than semantic meaning. The finding does not appear to be surprising or novel.
- Further analysis would be beneficial: show example outputs from each judge model and identify common errors; In Appendix I, where the authors justify the sample size, adding a power analysis would be ideal.

References:

[1] [Evaluating Large Language Models at Evaluating Instruction Following](https://arxiv.org/pdf/2310.07641)
[2] [The NarrativeQA Reading Comprehension Challenge](https://arxiv.org/pdf/1712.07040)

**Questions:**

- The error analysis in Table 2 shows judges struggle with under-specified answers. Could you provide examples of or qualitatively explain the under-specified answers that fooled even the best judges?

---

> ### Author Response · Authors · 2024-11-26
>
> We sincerely appreciate the reviewer's thorough assessment. We address the reviewer’s concerns about the weaknesses of the paper below.
>
> **1. Limited Scope:** Our focus on TriviaQA is deliberate and advantageous, establishing a clear baseline for LLM judging ability with minimal ambiguity. TriviaQA's multiple references per question provide a more robust ground truth, and our inter-annotator agreement (98% on 600 instances) surpasses LLMBar's (94% on 80 instances). Our larger sample size offers greater statistical power and scope, while setting a clear upper bound for human alignment in factual Q&A tasks. While additional datasets may be valuable for future work, our current methodology provides a focused and reliable assessment of LLM judging capability in academic and industrial research.
>
> **2. Over strict evaluation setup:** Our use of exact wording matches over semantic evaluation was intentional, reflecting real-world industry applications of Judge LLMs. In our preliminary research, we saw that semantic evaluation may lead to higher Judge scores but reduced human alignment. This methodology also allows for more consistent comparisons across different models and tasks, establishing a solid baseline for future research in this area.
>
> **3. Further analysis:** While a power analysis on the evaluation sample would be ideal, we’ve mentioned high statistical reliability of sample in Appendix I and performed error analysis on the evaluation sample, categorizing common errors into 5 groups (Section 5.2). Even the best Judge LLMs frequently provided completely incorrect or under-specified (partially incorrect) responses. Examples using GPT-4 and gold references are provided below. After implementing guidelines (Appendix G), below answers were identified as under-specified. Outside of these examples, most of GPT-4's errors were due to incorrect entities or factual inaccuracies.
>
> > **Q -** Which language - originally a form of Anglo-Chinese jargon - was used by traders and businessmen on the China coast?
> >
> > **GPT4 -** Pidgin
> >
> > **Gold -** Pidgin English, English based Pidgins, English Pidgin
>
> > **Q -** In the USA, the Tav HaYosher is a certification mark offered to establishments that do what?
> >
> > **GPT4 -** Serve Kosher food
> >
> > **Gold -** Serve Kosher food and treat their workers fairly
>
> In light of the rebuttal comments, would you be willing to update your score? Thank you!

---

### Official Review · Reviewer_jhCB · 2024-11-05

**Soundness:** 2
**Presentation:** 3
**Contribution:** 2
**Rating:** 3
**Confidence:** 4

**Summary:**

This work provides an examination of LLM judges regarding their performance and vulnerabilities in a reference-based evaluation setting for QA tasks. Using human annotation as the gold standard, a series of judge models are evaluated. For evaluation metrics, the manuscript proposes using Scott’s $\pi$ instead of accuracy, highlighting it as a main finding. It also shows that while less capable judges perform poorly at the instance level, i.e., giving the same decision as the human annotators, they achieve higher correlation with humans at the system level, i.e., producing a ranking of evaluated models by aggregating the instance-level decisions. Further analyses are conducted on changes in recall and precision scores, sensitivity to prompts, robustness, and leniency bias.

**Strengths:**

1. The provided analysis regarding the LLM judges' sensitivity to prompts, error types, a lack of robustness, and the leniency bias are interesting and valuable to future studies.

2. The paper is well-written and the findings are clearly presented.

**Weaknesses:**

It appears that some of the main findings in this work are either not well-supported, may lack generalizability, or have been discussed in previous work.

1. Lack of generalizability: The task setting of the LLM judges selected in this work is reference-based evaluation of QA, which differs from the common application scenario where LLM judges evaluate various tasks without a gold reference (e.g., AlpacaEval, Arena Hard). Access to gold references makes the evaluation task significantly easier. Therefore, the findings in this work may not generalize well to more open-ended, general evaluation settings. While it is stated that this task setting was chosen to reduce human disagreement, there exists a related dataset, LLMBar [1], which can be used to perform a more general evaluation of LLM judges, achieving over 90% human agreement (evaluation accuracy).

2. The finding that automatic evaluation metrics (specifically LLM judges) have a higher correlation with human evaluations at the system level than at the instance level has already been identified and well-discussed in related work on evaluating automatic evaluation of natural language generation tasks [2][3][4][5]. For example, [5] shows that automatic metrics can achieve higher system-level correlation with humans when they evaluate more instances. Therefore, this finding itself is not a novel contribution.

3. The manuscript proposes using Scott’s $\pi$ instead of accuracy as the evaluation metric for LLM judges, claiming that it "appears to be better able to discriminate among various judge models." However, this claim is not well-supported, as the only evidence provided is that Scott’s $\pi$ yields scores with a wider numerical range than accuracy, which could potentially be achieved by trivially rescaling the range. Further examination is needed to verify this claim, such as by demonstrating that Scott’s $\pi$ offers greater statistical power, with tighter confidence intervals or a lower p-value in significance tests. Additionally, the notion of separability defined in Arena Hard [6] would be useful for comparing evaluation metrics.

4. The finding that the true negative rate (resulting in a lower recall score) falls quickly with less capable judges does not hold when the two lexical-similarity-based metrics, exact match (EM) and Contain, are excluded. In fact, all the small LLM-based judges achieve higher recall scores than precision scores. The observed low true negative rate/recall score of EM and Contain is expected, as these metrics rely on lexical similarity and are likely to mark an answer that is correct but lexically different from the reference answers as incorrect.


References

[1] Zeng, Zhiyuan "Evaluating large language models at evaluating instruction following." ICLR 2024

[2] The price of debiasing automatic metrics in natural language evalaution (Chaganty et al., ACL 2018)

[3] Re-evaluating Evaluation in Text Summarization (Bhandari et al., EMNLP 2020)

[4] A Statistical Analysis of Summarization Evaluation Metrics Using Resampling Methods (Deutsch et al., TACL 2021)

[5] Re-Examining System-Level Correlations of Automatic Summarization Evaluation Metrics (Deutsch et al., NAACL 2022)

[6] Li, Tianle, et al. "From Crowdsourced Data to High-Quality Benchmarks: Arena-Hard and BenchBuilder Pipeline." arXiv preprint arXiv:2406.11939 (2024).

**Questions:**

The description of the platform and recruitment process for human annotations is unclear. Who are the annotators (e.g., crowdworkers), and what are the recruitment criteria?

---

> ### Author Response · Authors · 2024-11-26
>
> We are grateful to the reviewer for their comprehensive evaluation and specific feedback. We address each point raised in the review individually.
>
> **1. Lack of generalizability:** The reviewer points out that our work is a reference-based QA evaluation task which might not generalize to open-ended QA without the references. We have tested it on other datasets such as Natural Questions, and we have found that due to ambiguous questions (example question from NQ dataset - “Where did the term liberal arts come from ?”), we achieved less than 90% alignment scores among human annotators and Human - Judge LLM which can be attributed to the ambiguity in the question and references.
>
>
> Our choice of setup aimed to minimize ambiguity and establish an upper bound for human-LLM alignment in judging responses. By using TriviaQA's factual questions with multiple references, we reduced the potential for errors due to ambiguous questions or references. The observed differences in agreement between judges and humans in this sterile setup suggest that more complex scenarios, like open-ended questions without gold references, may present even greater challenges. Research conducted in the MT-Bench paper [1] also identified GPT-4 as the top-performing Judge LLM, but noted that its human alignment on the LLM evaluations was limited to ~81%. In this scenario, it’s then extremely hard to optimize Judge LLMs, when the dataset itself is so ambiguous that humans can’t agree to a correct answer. While LLMBar [2] achieved 94% agreement rate on a sample size of 80 instances between 2 annotators, we achieved 98% percentage agreement and 96% average alignment (Scott’s Pi) on a sample size of 600 between 3 annotators, which is a more reliable setup for our evaluation of Judge LLMs.
>
> **2. Novel contribution of Judge LLM correlation with human eval:** To our knowledge, this is the first study that is using Scott's pi to assess agreement between Judge LLMs across a wide range of model sizes (2B to GPT-4) and families. While existing work appreciates the performance of SOTA Judge LLMs, our work uniquely reveals the surprisingly strong performance of smaller models like Llama3.1-8B and Mistral-7B compared to larger ones. These findings could encourage more research into compact models for various applications, benefiting the research community.
>
> **3. Scott’s pi claims are not well supported:** Scott's π is preferred not only for its broader numerical scope but primarily for its ability to assess inter-judge agreement beyond coincidence. This offers a more sophisticated analysis of judges' evaluation performance [Appendix B] and has been explored in literature as well [3]. By introducing Scott's π, we aim to promote a more accurate measure of inter-rater reliability across various fields. The statistical reliability of Scott's π has been explored in literature [4] [5], and Appendix I discusses the reliability of evaluation samples through consistent Scott's π scores.
>
> **4. True Negative rate of Judge LLMs:** We appreciate your observation! Indeed, EM/Contains Judge LLMs show higher recall than precision as expected. We initially included these to demonstrate lexical metrics' limitations in assessing semantic richness, despite their alignment with some Judge LLMs. We'll update the paper to mention this in the text and remove EM/Contains from the Precision/Recall graph for clarity.
>
> **5. Information about annotators and recruitment criteria:** Preliminary research involved iterative refinement of human annotation guidelines [Appendix G] to ensure consistency and reproducibility across annotators with general English semantic knowledge. CS graduate students served as annotators for this experiment.
>
> In light of the rebuttal comments, would you be willing to update your score? Thank you!
>
> **References -**
>
> [1] Judging LLM-as-a-Judge with MT-Bench and Chatbot Arena (Zheng et al., NeurIPS 2023)
>
> [2] Evaluating Large Language Models at Evaluating Instruction Following (Zeng et al., ICRL 2024)
>
> [3] Interrater reliability estimators tested against true interrater reliabilities (Zhao, PMCMed Central, 2022)
>
> [4] Answering the Call for a Standard Reliability Measure for Coding Data (Hayes et al., Communication Methods and Measures 2007)
>
> [5] A Critical Analysis of Inter-Coder Reliability Methods in Information Systems Research (Nili et al., ACIS 2017)

---

### Note · Authors · 2024-12-15

**Comment:**

Thank you for the reviewers' detailed feedback. As of now, we have not received responses to 4 out of 5 of our rebuttals, and our score stands at 4.8. The one rebuttal we did receive feedback on resulted in an increased score. However, due to the lack of responses from the remaining reviewers, we have decided to respectfully withdraw our paper from this conference.

Thank you.

**Withdrawal Confirmation:**

I have read and agree with the venue's withdrawal policy on behalf of myself and my co-authors.